

# Contrasting decadal trends of subsurface excess nitrate in the western and eastern North Atlantic Ocean

Jin-Yu Terence Yang[1,2], Kitack Lee[1]*, Jia-Zhong Zhang[3], Ji-Young Moon[1],

Joon-Soo Lee[4], In-Seong Han[4], and Eunil Lee[5]

[1] Division of Environmental Sciences and Engineering, Pohang University of Science and Technology, Pohang 37673, Korea.

[2] State Key Laboratory of Marine Environmental Science, College of Ocean and Earth Sciences, Xiamen University, Xiamen 361102, China.

[3] National Oceanic and Atmospheric Administration, Atlantic Oceanographic and Meteorological Laboratory, Miami, FL 33149, USA.

[4] Ocean Climate and Ecology Research Division, National Institute of Fisheries Science, Busan 46083, Korea.

[5] Ocean Research Division, Korea Hydrographic and Oceanographic Agency, Busan 49111, Korea.

**\*Corresponding author,**
Kitack Lee, Professor
Division of Environmental Sciences and Engineering
Pohang University of Science and Technology
Phone: +82-54-2792285; fax: +82-54-2798299; e-mail: ktl@postech.ac.kr

**Email for other co-authors :**
Jin-Yu Terence Yang (jyyang@xmu.edu.cn); Jia-Zhong Zhang (jia-zhong.zhang@noaa.gov); Ji-Young Moon (todaud@postech.ac.kr); Joon-Soo Lee (leejoonsoo@korea.kr); In-Seong Han (hisjamstec@korea.kr); Eunil Lee (elee@korea.kr)

**Keywords,**

anthropogenic nitrogen; excess nitrate; North Atlantic Ocean; climate variability; decadal trend





## Abstract

Temporal variations in excess nitrate ($DIN_{xs}$) relative to phosphate were evaluated using datasets derived from repeated measurements along meridional and zonal transects in the upper (200–600 m) North Atlantic (NAtl) between the 1980s and 2010s. The analysis revealed that the $DIN_{xs}$ trend in the western NAtl differed from that in the eastern NAtl. In the western NAtl, which has been subject to atmospheric nitrogen deposition (AND) from the USA, the subsurface $DIN_{xs}$ concentrations have increased over the last two decades. This increase was associated with the increase in AND measured along the US east coast, with a mean lag period of 15 years. This time lag was approximately equivalent to the time elapsed since the subsurface waters in the western NAtl were last in contact with the atmosphere (the ventilation age). Our finding provides an evidence that the $DIN_{xs}$ dynamics in the western NAtl in recent years has been affected by anthropogenic nitrogen inputs, although this influence is weak relative to that in the North Pacific. In contrast, a decreasing trend in subsurface $DIN_{xs}$ was observed after the 2000s in the eastern NAtl, particularly in the high latitudes. This finding may be associated with a possible decline of tropical $N_2$ fixation and the weakening of the Atlantic meridional overturning circulation, although more time-resolved data on nutrients and meridional circulation are needed to assess this hypothesis. Our results highlight the importance of both anthropogenic and climate forcing in impacting the nutrient dynamics in the upper NAtl.



## 1. Introduction


The supply of reactive inorganic nitrogen ($Nr = NO_y + NH_x$) to the surface ocean is limited
in most of the oligotrophic marine environments (Fanning, 1989; Moore et al., 2013; Moore,
2016). The addition of Nr will lead to an increase in primary and export production, and
eventually an enhancement of carbon sequestration in the ocean interior (Okin et al., 2011;
Jickells and Moore, 2015). Anthropogenic nitrogen deposition (AND) to the contemporary ocean
is comparable in magnitude to marine biological $N_2$ fixation, which has been thought to be the
major external source of Nr to the oligotrophic ocean (Duce et al., 2008; Fowler et al., 2013;
Jickells et al., 2017). In particular, AND has been found to enhance phytoplankton productivity
in Nr-depleted tropical and subtropical oceans located downwind of continents that are sources
of pollutant nitrogen (Kim et al., 2014b; St-Laurent et al., 2017). Any changes in this external
source of Nr induced largely by human activities could cause a wide range of ecological and
biogeochemical consequences (e.g., Doney et al., 2007; Yang et al., 2016).
The impact of AND on the dissolved inorganic nitrogen concentration (DIN) in seawater
has recently been assessed in coastal and marginal seas (Kim et al., 2011; Moon et al., 2016), and
in the remote open ocean (Kim et al., 2014a), using historical nutrient concentration datasets.
The analysis of 30 years of data showed that the DIN has increased in marginal seas off the
northeast Asian continent, whereas the dissolved inorganic phosphorus concentration (DIP) has
remained relatively unchanged over this time period (Kim et al., 2011). For open-ocean areas,
the temporal change in excess nitrate relative to DIP ($DIN_{xs} = DIN - R_{N:P} \times DIP$, where $R_{N:P}$ is
the average DIN:DIP ratio of 15:1 for deep waters) was estimated using the relationship between
$DIN_{xs}$ in a particular water parcel and the chlorofluorocarbon (CFC)-12-derived ventilation age
of that parcel (Kim et al., 2014a). An underlying assumption in this analysis is that ocean



biological processes (i.e., production and microbial oxidation of organic matter) operate at a
DIN:DIP ratio of 15:1 and thus do not change $DIN_{xs}$. Changes in seawater $DIN_{xs}$ only occur
when either N input (i.e., AND and $N_2$ fixation) or N loss (i.e., denitrification) occurs. The
analysis using this method revealed that the $DIN_{xs}$ has increased in the western North Pacific
Ocean (NPO) since the 1970s (Kim et al., 2014a).

The addition of Nr to the North Atlantic Ocean (NAtl), which is located downwind from

North America, has more than doubled since 1986 (Galloway et al., 1996). The increasing
addition of Nr may lead to an increase in $DIN_{xs}$ in the NAtl (Zamora et al., 2010). However, it
has been argued that $N_2$ fixation is a more likely cause of the higher subsurface $DIN_{xs}$ in the
NAtl (Gruber and Sarmiento, 1997; Bates and Hansell, 2004). Differentiating the contributions
of $N_2$ fixation and AND is not straightforward because both processes leave similar
biogeochemical signals in seawater, including a high DIN:DIP ratio and low nitrogen isotope
composition (Hastings et al., 2003; Knapp et al., 2010; Yang et al., 2014). In addition to these
two processes, climate variations (commonly expressed as the North Atlantic Oscillation Index)
can concurrently influence the nutrient dynamics in the NAtl (Bates and Hansell, 2004; Singh et
al., 2013). As a result, the processes causing the change in the subsurface $DIN_{xs}$ signal in the
NAtl remain unresolved. This knowledge gap needs to be filled to improve understanding of the
marine nitrogen cycle (Gruber, 2008).

The present study was designed to explore the occurrence and rate of decadal change in

$DIN_{xs}$ ($\Delta DIN_{xs}$ in $\mu$mol kg$^{-1}$ decade$^{-1}$) in the subsurface NAtl, as well as the explanations for
$\Delta DIN_{xs}$, based on repeat measurements of nutrients and other oceanographic parameters made
over the past three decades or longer.





## 2. Materials and methods

### 2.1. Data

Historical data on temperature, salinity, and the concentrations of nitrate, nitrite, phosphate and oxygen used in this study have been collected as parts of the Transient Tracers in the Ocean (TTO), the World Ocean Circulation Experiment (WOCE), the Climate Variability $CO_2$ Repeat Hydrography (CLIVAR), and the Global Ocean Ship-Based Hydrographic Investigations (GO-SHIP) programs. Analysis of nutrient data was based only on concentrations greater than 0.1 $\mu$mol kg$^{-1}$ for DIN and 0.01 $\mu$mol kg$^{-1}$ for DIP. These concentration levels approximate the detection limits of DIN and DIP for the analytical methods used in the field observations (Zhang et al., 2001; Hydes et al., 2010). The data used in our analysis are available at https://www.nodc.noaa.gov/ocads/oceans/ (the Global Ocean Data Analysis Project Version 2, GLODAPv2 product and CLIVAR database).

Data analysis was primarily focused on three meridional (A22, A20, and A16N) transects in the NAtl (Fig. 1). A zonal transect (A05) was also included for comparison. All four transects used in the study are located downwind of the North American continent, which is a major source region of anthropogenic nitrogen (Fig. 1). Each of these transects was sampled 3 or 4 times during the past 30 years (Table S1). To extend temporal data coverage in the analysis, historical data obtained from locations adjacent to the four study transects were included in the analysis. Moreover, the repeat measurements along transect A22 occurred on slightly different tracks, particularly in the Caribbean Sea and in the northern end of the transect. We therefore excluded data for areas south of Puerto Rico (~18.5°N) and north of 36°N, where the distance between the location of repeat measurements exceeded 2° longitude. Data obtained south of 20°N along A16N and south of 15°N along A20 were also excluded from the analysis, because



these regions are considerably influenced by the water masses originated from the equatorial
upwelling region (Hansell et al., 2004), and any change in the intensity of upwelling could bias
our analysis of changes in $DIN_{xs}$. To make datasets collected in different years consistent, an
inverse analysis of repeated measurements made at the same locations was performed to estimate
measurement biases. Any biases found were accounted for by applying adjustment factors to the
original datasets. The adjusted datasets were reported in the GLODAPv2 product and CLIVAR
database (Key et al., 2015; Olsen et al., 2016).

The datasets from the GLODAPv2 and CLIVAR did not have systematic biases (Table

S1). To account for any small discrepancies that may exist among the various datasets collected
on the 4 transects, we adjusted the DIN and DIP concentrations based on the assumption that the
physical and chemical properties in deep waters of the tropical and subtropical NAtl (south of
50°N) did not change on decadal time scales (Figs. S1 and S2; see details in Text S1). The mean
corrections were found to be $0.04 \pm 0.03$ $\mu$mol kg$^{-1}$ for DIN and $0.006 \pm 0.004$ $\mu$mol kg$^{-1}$ for
DIP, corresponding to their adjustment factors mostly less than 1.5% (Table S2 and Fig. S3).
These corrections fell within the detection limits for DIN and DIP, and were an order of
magnitude smaller than the subsurface $\Delta DIN_{xs}$ signals (see section 3.1). The finding that the
subsurface $\Delta DIN_{xs}$ signals were considerably greater than the detection limit of DIN is a strong
indication that our data adjustments probably did not influence the temporal trend of $DIN_{xs}$. It
also suggests that our method can extract the decadal trends of $DIN_{xs}$ from less time-resolved
datasets, as has successfully been used in previous studies (Zhang et al., 2000; Ríos et al., 2015;
Woosley et al., 2016).
**2.2. Relative abundance of DIN over DIP ($DIN_{xs}$) in water parcels**





We calculated the DIN surplus or deficit relative to DIP in each seawater sample (i.e. the
deviation of the DIN:DIP ratio from that in deep water) by calculating $DIN_{xs}$ (Fig. 2). This
calculation was performed in the upper 1500 m; and the $\Delta DIN_{xs}$ signals between GO-SHIP and
WOCE time periods were also evaluated using data obtained from the subsurface layer (200–600
m) because the majority of $DIN_{xs}$ signals derive from this layer and hence any changes would be
expected to be more marked (see the next section and the $\Delta DIN_{xs}$ signals at 1200–1500 m for
reference in Fig. 2). In addition, the effect of seasonal variations on $DIN_{xs}$ signals at this depth
layer is generally insignificant, because seasonal variations are largely confined to waters
shallower than the climatological winter mixed layer (down to 200 m depth). Based on analysis
of data obtained from the BATS site, seasonal variations in subsurface (the target water depth
range) mean $DIN_{xs}$ values were $< 0.1$ $\mu$mol kg$^{-1}$ (Note that nutrient data at BATS are available at
http://batsftp.bios.edu/ BATS/bottle/bats_bottle.txt).
The values for $\Delta DIN_{xs}$ and nutrients in the water column could be biased because of
mixing of water masses having different $DIN_{xs}$ concentrations, and different nitrogen to
phosphorus ratios associated with organic matter oxidation during various observation periods.
To minimize biases caused by these natural processes, we examined changes in potential
temperature $\theta$, salinity and AOU along the potential density surfaces $\sigma_\theta$ (corresponding to 200–
600 m where the $DIN_{xs}$ signals are the largest) at 5°–15° latitude or longitude intervals
representing average regional variations along each transect. We found that the $\theta$ and salinity of
a water mass occupying any given density surface did not change between repeat occupations (p
$> 0.05$, Student's t-test and ANOVA with Games-Howell test), except for slightly lower $\theta$ and
salinity since 2000s in the subpolar region (north of 45°N) along A16N (Figs. S4 and S5). This
finding is a strong indication that biases in $\Delta DIN_{xs}$ from the mixing of different water masses

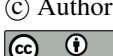



were negligible in the subtropical regions over the observation periods (approximately 30 years).
In contrast, we found slight differences in $\Delta$AOU among a few of selected reoccupations (not
shown). To remove the contribution of DIN and DIP from oxidation of organic matter,
adjustment of the nutrient concentrations was made by using the DIP:DIN:$O_2$ remineralization
ratio of 1:15:(−160) derived from data along the $\sigma_\theta = 27.0$ horizon in the NAtl (Takahashi et al.,
1985; Anderson and Sarmiento, 1994). The estimated DIN:DIP ratio for remineralization of
organic matter was in the range 15–18 (Takahashi et al., 1985; Anderson and Sarmiento, 1994).
The chosen value of the DIN:DIP ratio for remineralization did not significantly change the
patterns of $\Delta$DIN$_{xs}$. Therefore, the $\Delta$DIN$_{xs}$ within the layer of the DIN$_{xs}$ maxima along all
transects examined were free from biases of either mixing of water masses or changes in
oxidation of organic matter. For each subregion, DIN$_{xs}$ (or DIP) anomalies indicated individual
DIN$_{xs}$ (or DIP) values minus the mean DIN$_{xs}$ (or DIP) values from the GO-SHIP dataset (Figs.
3–5). Note that the positive anomalies indicate higher values than the GO-SHIP data.

## 182    3. Results and discussion

### 183    3.1. Decadal variations of DIN$_{xs}$ in the upper North Atlantic Ocean

High DIN$_{xs}$ values were broadly distributed in the subsurface waters (< 1000 m) in the

NAtl. In particular, the maximum DIN$_{xs}$ values were found between 200 and 600 m (Fig. 2), and
were slightly higher in the western basin (an average value of $1.4 \pm 0.3$ $\mu$mol kg$^{-1}$ was calculated
for A22 in 2012) than those in the eastern basin (an average value of $0.8 \pm 0.2$ $\mu$mol kg$^{-1}$ was
calculated for A16N in 2013). Similarly, the west-east gradient in DIN$_{xs}$ was apparent along the
zonal transect A05 at 24.5°N latitude (Fig. S6).



Based on multiple cruises along each transect, changes in $DIN_{xs}$ were discernable over the
decadal periods; these changes were most pronounced between 200 m and 600 m (Fig. 2). The
rate of $\Delta DIN_{xs}$ in the NAtl differed among locations of transects between GO-SHIP and WOCE
periods. Specifically, the $\Delta DIN_{xs}$ values were mostly positive in the western NAtl (A22 and
A20), where they varied from 0.02 to 0.33 $\mu$mol kg$^{-1}$ decade$^{-1}$, with the highest rate found at
31°N–36°N along A22. In contrast, the $\Delta DIN_{xs}$ values became negative in the eastern NAtl
(20°N–60°N along A16N), where they ranged from –0.07 to –0.40 $\mu$mol kg$^{-1}$ decade$^{-1}$; the
greatest rate of $DIN_{xs}$ decrease was in the subpolar region (north of 45°N). Moreover, the $\Delta DIN_{xs}$
values remained close to zero in the intermediate waters (1200–1500 m) in the western and
eastern subtropical NAtl (Fig. 2). This observation confirms that the marked changes in $DIN_{xs}$
largely occurred in the subsurface waters.
The $\Delta DIN_{xs}$ in the NAtl showed geographically distinct patterns after removing the
influence of remineralization of organic matter (Fig. 3). We found that the $\Delta DIN_{xs}$ within the
layer of the $DIN_{xs}$ maximum decreased since 1997 (measurement year) along the transects near
the source continent (i.e. the entire transect of A22, and 31°N–36°N along A20) (Figs. 3a and
3b), and its rates ranged from 0.19 to 0.33 $\mu$mol kg$^{-1}$ decade$^{-1}$ (Fig. 2). The trend in $DIN_{xs}$ found
at the BATS site is broadly comparable to that found between 31°N and 36°N along A22 (Fig.
3a). The rate of increase of $DIN_{xs}$ at the BATS site since the late 1990s (0.40 $\mu$mol kg$^{-1}$ decade$^{-}$
$^{1}$) was also similar to that observed in the latitude band 31°N–36°N along A22. Such agreement
with time series data strengthens our finding derived from less time-resolved datasets.
The discernable increase in $DIN_{xs}$ rapidly diminished in the central gyre of the NAtl
(15°N–31°N for A20, 20°N–45°N for A16N and A05), where the $\Delta DIN_{xs}$ was statistically
insignificant ($p > 0.05$, Student's t-test and ANOVA with Games-Howell test; Figs. 3 and S7).





Furthermore, the level of $DIN_{xs}$ appeared to decrease at high latitudes in the eastern NAtl (north
of 45°N on A16N; Fig. 3c). The trend of decrease has been more pronounced since the 2000s in
this region, and occurred concurrently with decreases in temperature and salinity ($p < 0.05$,
Student's t-test and ANOVA with Games-Howell test; Figs. S4c and S5c). Our observations
indicate that the mechanisms responsible for the $\Delta DIN_{xs}$ in the subtropical and subpolar NAtl are
likely to differ.
**3.2 AND influence on the $\Delta DIN_{xs}$ in the western North Atlantic Ocean**
More pronounced increase in the subsurface $DIN_{xs}$ has been observed in recent decades in
the western mid-latitude NAtl (Fig. 3), which is subject to considerable AND input from the
North American continent. Recent studies suggest that the reduced form of nitrogen entering the
NAtl is primarily of marine autochthonous origin, rather than of anthropogenic origin (i.e.
atmospheric deposition) (Altieri et al., 2014; Altieri et al., 2016). Thus, this autochthonous
reduced nitrogen would not influence seawater $DIN_{xs}$ values. Therefore, we only included the
effect of $NO_x$ emissions and deposition (mostly in the forms of $NO_3^-$ and $HNO_3$; hereinafter $NO_x$
is used to represent the major form of AND) on the $\Delta DIN_{xs}$ values (Figs. 1 and 5b). From the
1970s to the 2010s the $NO_x$ emissions from the USA showed a three-phase temporal transition
(EPA, 2000). The $NO_x$ emissions from the USA increased from 1970 to the mid 1980s, and
stayed at high levels for approximately 20 years, and then decreased gradually after the mid
2000s as a result of the regulation of air pollutant emissions throughout the North American
continent (Fig. 5b). The anthropogenic nitrogen pollutants are mostly transported eastward and
ultimately deposited in the western NAtl (Fig. 1).
Although there are limited data (time and space) on wet deposition of $NO_x$, the temporal
pattern based on measurements on the US Atlantic coast is comparable to that for the $NO_x$



emissions. For example, based on data obtained from the National Atmospheric Deposition
Program (NADP; http://nadp.sws.uiuc.edu/), there was an increase from the 1980s to early
1990s, and the level remained high for approximately 15 years and then decreased (Fig. 5b). This
trend of wet deposition of $NO_x$ was commonly found at AND monitoring sites located along the
US Atlantic coast (Table S3).

The AND signals can be transported to the subsurface waters of the mesopelagic western

NAtl via two associated mechanisms. The first process involves production and bacterial
oxidation of organic matter. In these biological processes, new anthropogenic nitrogen added by
atmospheric deposition are removed from the surface via photosynthetic utilization by
phytoplankton and gravitational sinking of the resulting organic matter with N:P ratio higher
than 15:1 (Singh et al., 2013). These N-rich organic matters are subsequently remineralized by
bacteria at depth (Antia, 2005). This process involving well-known biological processes would
facilitate the transfer of high surface N:P signals to the subsurface waters. The second process
involves the physical transport of surface waters with greater N:P signals, which is a plausible
mechanism for generating the subsurface AND signals observed in the western NAtl. High
inputs of $NO_x$ by atmospheric deposition occur over the coastal areas of the NAtl and are mostly
entrained in areas close to the northern edge of the western NAtl via the strong and persistent
western boundary current (i.e. the Gulf Stream, Fig. 1). Both active winter mixing and the
concurrent formation of mode water in this region would be expected to facilitate the transport of
surface waters loaded with high $DIN_{xs}$ (and anthropogenic $CO_2$ and CFCs) to the subsurface
layer, and to spread these $DIN_{xs}$-loaded waters southward (Palter et al., 2005).

The substantial increase in subsurface $DIN_{xs}$ after 1997 (approximately equal to the pCFC-

12 ventilation year of 1982) at sites having greater inputs of AND (boxes 1–3 in Fig. 5a, and at



the BATS site) appears to coincide with the increasing wet deposition of $NO_x$ from the US
continent, with a lag period of approximately 15 years. The time lag observed is approximately
equal to the ventilation age of the target subsurface waters in this region, which was estimated to
be 6–25 years based on the CFCs concentrations (Hansell et al., 2004; Hansell et al., 2007). The
time lag suggests that the physical mechanism is important in transporting the AND signals to
the subsurface waters, although the mismatch between the observed time lag and the ventilation
age of water masses may be due, at least in part, to the biological processes.

The rates of $DIN_{xs}$ increase (0.19–0.33 $\mu$mol kg$^{-1}$ decade$^{-1}$; boxes 1–3 in Fig. 5a) measured

in the western NAtl (reported in the preceding section) are equivalent to an increase of 78–135
mmol N m$^{-2}$ decade$^{-1}$ in the subsurface N inventory (200–600 m) of the western NAtl. This is
slightly higher than the increase in wet $NO_x$ deposition (approximately 60 mmol N m$^{-2}$ decade$^{-1}$)
measured along the US east coast from the 1980s to the 2000s (Fig. 5b), but is broadly consistent
with the total $NO_x$ fluxes (approximately 90 mmol N m$^{-2}$ decade$^{-1}$) if dry deposition is included
in the modeled and observed results (Dentener et al., 2006; Baker et al., 2010). We thus suggest
that anthropogenic nitrogen input is probably a main driver of $DIN_{xs}$ increase in the western
basin. An anthropogenic influence manifested in oceanic nutrient dynamics having a lag period
of 15 years, has also been detected at 200–600 m in the Mediterranean Sea, where Moon et al.,
(2016) showed a three-phase temporal transition (a trend of increase–stability–decline) in DIN
concentration between 1985 and 2014; this was probably associated with corresponding changes
in anthropogenic nitrogen input.

The temporal trend of the nitrogen isotope record (CS-$\delta^{15}$N) measured on the Bermuda

coral skeleton is comparable to the trends of $NO_x$ emission from the USA (Fig. S8), indicating
that the AND signals have been embedded in the coral $\delta^{15}$N record. The CS-$\delta^{15}$N record on the





Bermuda coral reflects the annual biological response to the local AND signals in the surface
waters; hence, its trend may follow changes in anthropogenic $NO_x$ input without a time lag. For
the western NAtl, the rates of $DIN_{xs}$ increase we found are in agreement with those from the
earlier studies using different datasets and methodologies (Hansell et al., 2007; Landolfi et al.,
2008; Singh et al., 2013), but are lower than those observed in the NPO (0.30–1.20 $\mu$mol kg$^{-1}$
decade$^{-1}$, Kim et al., 2014a). The different rates of seawater $DIN_{xs}$ increase found between the
western NAtl and NPO appear to be consistent with the CS-$\delta^{15}$N records in these two basins.
During the 20$^{th}$ century, a small decline (–0.2‰) in CS-$\delta^{15}$N was observed in corals from
Bermuda (Wang et al., 2018), whereas a greater decrease (–0.7‰) in CS-$\delta^{15}$N was detected from
the South China Sea (Ren et al., 2017). The lower rates of seawater $DIN_{xs}$ increase (or slower
decline in CS-$\delta^{15}$N) in the NAtl were likely due to the lower rate of nitrogen emissions (also
indicating nitrogen deposition) from the North American continent (0.15 Tg N year$^{-1}$ observed
from 1970 to 2000; EPA, 2000) than from northeast Asia (0.40 Tg N year$^{-1}$ observed from 1980
to 2010; Liu et al., 2013). In this case, the recent trend of decreasing emission in anthropogenic
nitrogen from North America, as well as the decrease in wet nitrogen deposition observed along
the US east coast, may reverse the pattern of the increase in subsurface $DIN_{xs}$ in the western
NAtl in the near future. Indeed, this reversed pattern appears to have emerged recently at the
BATS site (Figs. 3a and 5b), based on more time-resolved observations. Together, our findings
suggest that the AND has affected the nutrient dynamics in the western NAtl, although the
magnitude of this effect is relatively small, and its influence would be expected to become less
significant under a scenario of increased control of pollutant emissions.
**3.3. Biogeochemical processes that may affect the $\Delta DIN_{xs}$ in the western North Atlantic**
**Ocean**



Other biogeochemical processes may also affect the observed pattern of $\Delta DIN_{xs}$ in the
western NAtl. Nitrogen fixation contributes considerably to the total export production (1.3–3.8
mol C m$^{-2}$ year$^{-1}$; Lee, 2001) in oligotrophic gyres of the NAtl (Lee et al., 2002; Ko et al., 2018),
which could therefore generate the positive signals of $DIN_{xs}$ in subsurface waters (Hansell et al.,
2004). The rate of $N_2$ fixation and the abundance of diazotrophs have been reported to be highest
(Luo et al., 2012; Benavides and Voss, 2015) in the subtropical gyre of the western NAtl (see
boxes 4–6 in Fig. 5a), however, the subsurface $DIN_{xs}$ did not change significantly among the
repeat occupations of transects (Figs. 3 and S7a). No direct evidence for increasing activity of
diazotrophs in the NAtl is available (Mahaffey et al., 2005; Benavides and Voss, 2015). Contrary
to our expectation, the increase in subsurface $DIN_{xs}$ was only found upstream of the subduction
zone (north of the hot spots for $N_2$ fixation; Figs. 3 and 5a). In this region (boxes 1–3) the
observed rate of $N_2$ fixation was 4.2 mmol m$^{-2}$ year$^{-1}$ (Luo et al., 2012), considerably lower than
the atmospheric $NO_x$ deposition (10–40 mmol m$^{-2}$ year$^{-1}$; see Fig. 1). If $N_2$ fixation mainly
drives the increase in subsurface $DIN_{xs}$ in this region, its rate would have been expected to
increase by 2 to 3-fold during recent decades. Such an increase in $N_2$ fixation activity is highly
unlikely (Benavides and Voss, 2015). Moreover, if $N_2$ fixation activity had increased during the
study period we would expect the DIP concentration decreases in the surface ocean and
concurrently increases below (Kim et al., 2014a), but this was not observed (Fig. 4). Therefore,
$N_2$ fixation has probably not been a major factor leading to the increase in $DIN_{xs}$ in the western
NAtl over the study period.
Remineralization of particulate and dissolved organic matters (POM and DOM) is another
potential source of subsurface $DIN_{xs}$ in the NAtl, as a result of the high N:P ratios of organic
matters and the preferential remineralization of P from POM and DOM (Landolfi et al., 2008;





Lomas et al., 2010). The DON concentration in the subsurface waters in the western NAtl (near
the BATS site), however, remained unchanged during the period 1998–2011 (http://bats.bios.
edu/). Moreover, the N:P ratios in DOM and suspended POM obtained at 0–100 m at the BATS
site did not change between 2004 and 2012 (Singh et al., 2015). Likewise, we did not find any
discernible interannual changes in the N:P ratio of sinking particles collected between 150 and
300 m at the BATS site (Fig. S9). Thus, the change in subsurface $DIN_{xs}$ in the western NAtl was
not primarily driven by variable N:P ratios of sinking POM. Taken together, these findings
suggest that DOM and POM remineralization has not contributed to the $\Delta DIN_{xs}$ in the subsurface
waters of the western NAtl during the periods in this study. Having excluded $N_2$ fixation and
remineralization of organic matters as key drivers, we hypothesize that the addition of AND has
been the major contributor to the recent increases in subsurface $DIN_{xs}$ in the western NAtl.
**3.4. Influences of climate variability on the $\Delta DIN_{xs}$ in the western North Atlantic Ocean**

As a prevailing climate mode over the NAtl, the North Atlantic Oscillation (NAO) strongly

influences the formation of the subtropical mode water (STMW) in the western NAtl, which in
turn affects subsurface nutrient and $DIN_{xs}$ concentrations in the downstream region (Bates and
Hansell, 2004; Palter et al., 2005). The STMW is known to form in areas south of the Gulf
Stream extension, and then primarily flows southward to the entire western basin; its intrusion to
the eastern basin has been suggested to be minor (Palter et al., 2005; Palter et al., 2011). The
formation of the STMW is generally enhanced when the NAO index becomes negative. During
this negative phase of the NAO, an increased contribution of low-nutrient water to the STMW
lowers the subsurface nutrient concentrations and $DIN_{xs}$ in the subtropical gyre. In contrast,
during the positive phase of the NAO, the STMW formation becomes weaker, and thus the
subsurface nutrient concentrations and $DIN_{xs}$ would increase downstream of the STMW





formation region (Palter et al., 2005). The winter (December–March) NAO index has been
mostly positive values since 1980, although its trend appeared to show an increase before the
early 1990s and to decrease slightly thereafter (Fig. S10). Contrary to the trend in this
atmospheric forcing, our nutrient data showed no evident changes in the subsurface DIP in the
downstream region (e.g. A22 and A20; $p > 0.05$, Student's t-test and ANOVA with Games-
Howell test; Figs. 4 and S7b) over the past 3 decades, irrespective of changes in the NAO index.
These observations indicate that the basin-wide $\Delta DIN_{xs}$ are probably less likely controlled by a
persistent positive phase of the NAO. Time-series data further strengthened the conclusion
drawn from the basin-scale data. For example, the decline in the Bermuda CS-$\delta^{15}$N value was
accompanied by several superimposed decadal oscillations induced by the NAO (Wang et al.,
2018). Similarly, such oscillations appear to be imprinted in the time-series measurements of
subsurface $DIN_{xs}$ at the BATS site (Fig. 5b). Nonetheless, the basin-wide $\Delta DIN_{xs}$ trends induced
by anthropogenic inputs of nitrogen are still visible.
**3.5. Subsurface $\Delta DIN_{xs}$ trend in the eastern North Atlantic Ocean**

There was an apparent decrease in subsurface $DIN_{xs}$ in the eastern NAtl (e.g. A16N),

which is the opposite trend to that found in the western NAtl (Fig. 5a). The decreasing trend (−
0.40 $\mu$mol kg$^{-1}$ decade$^{-1}$) in the subsurface $DIN_{xs}$ in the eastern subpolar NAtl (45°N–60°N
along A16N) has been more evident since the 2000s (Fig. 3c). A similar decrease in the
subsurface (300–500 m) DIN between 1998 and 2013 was also found at a site (68.0°N, 12.7°W)
in the northern Iceland Sea, but little change in DIP was observed (Fig. S12). The
remineralization of DOM may play an important role (Kähler et al., 2010); however, due to
insufficient time-resolved data, we could not confirm whether there was any decrease in the rate
of DOM remineralization in the eastern subpolar NAtl. From 1999 to 2009, NO$_x$ emission from



Europe has decreased by 31%, mainly owing to change in energy consumption from fossil fuels
to nuclear power (Vet et al., 2014). This decline in recent $NO_x$ emission from Europe (the blue
solid line in Fig. S11) is a viable explanation for the decrease in subsurface $DIN_{xs}$ in the eastern
subtropics. However, much of the $NO_x$ derived from Europe is probably deposited to the
European coasts, because the prevailing westerly winds carry it eastward to the eastern European
continent (Fig. 1) (Baker et al., 2010). Moreover, the amounts of $NO_x$ deposited onto the eastern
subpolar basin ($< 10$ mmol N m$^{-2}$ year$^{-1}$) were found to be small (Fig. 1). In the extreme scenario
in which no such $NO_x$ deposition occurred during the period of analysis, the lack of this $NO_x$
deposition would only account for $< 20\%$ of the total decline in subsurface $DIN_{xs}$ in the eastern
subpolar NAtl (Fig. 2c). This suggests that the influence of European AND on seawater nutrient
dynamics in the eastern subtropical NAtl is minimal (Hansell et al., 2007).

A recent study reported that most $N_2$ fixation occurs in the tropical NAtl (south of 24°N;

Marconi et al., 2017). The upper loop of the Atlantic meridional overturning circulation (AMOC)
acts as a passage through which nutrients are transported from the tropical regions to the
subpolar gyre (Pelegrí and Csanady, 1991; Williams et al., 2006; Moore et al., 2009). Therefore,
a decreasing rate of $N_2$ fixation is another likely explanation for the decreasing trend in
subsurface $DIN_{xs}$ in the eastern NAtl. Indeed, the supply of iron and phosphorus from Africa to
the eastern tropical and subtropical NAtl has declined by approximately 10% per decade during
the period 1980–2008 (Foltz and Mcphaden 2008; Ridley et al., 2014), and was likely
accompanied by a reduction in the growth of diazotrophs in the eastern tropical NAtl (Mills et
al., 2004; Benavides et al., 2013). On the other hand, the decrease in the subpolar subsurface
$DIN_{xs}$ occurred concurrently with decreases in temperature and salinity at the same water depth
in the subpolar region (Figs. 3c, S4c and S5c). These observations of concurrent cooling and





freshening were suggested to be associated with the fact that the strength of ocean circulation (as
well as heat transport) in the NAtl is decreasing, as a result of weakening of the AMOC since
2005 (Srokosz and Bryden, 2015; Robson et al., 2016). Therefore, the observed weakening in the
northward transport of tropical waters might in part contribute to a reduction in the $DIN_{xs}$
signature in the subpolar gyres of the eastern NAtl, as seen north of 45°N on A16N (Fig. 5a).
This result is consistent with a reduction in the subsurface $DIN_{xs}$ observed in the north Iceland
Sea since 2005 (Fig. S12d). If this mechanism plays a governing role, changes in nutrient
supplies related to a reduction of the AMOC would have a large impact on biological export and
the marine ecosystem (Schmittner, 2005). Although compelling, our observations are not a direct
confirmation that decline in $N_2$ fixation rate and variations in the AMOC are the main cause of
the decreasing trend in subsurface $DIN_{xs}$ in the eastern subpolar region. Therefore, this finding
remains a working hypothesis that needs confirmation using more time-resolved data in future
studies.

## 411    4. Conclusions and implication

Our results support that AND has been a cause of the temporal variations in seawater

$DIN_{xs}$ in the subsurface waters of the western NAtl during the recent 3 decades. In the eastern
subtropical NAtl, the decline in $NO_x$ emission from Europe, and possibly a decrease in the $N_2$
fixation rate, are likely drivers of subsurface $DIN_{xs}$ change. However, in the subpolar gyres of
the eastern NAtl the subsurface $DIN_{xs}$ has decreased since the 2000s, as a result of a possible
decrease in $N_2$ fixation. Recent changes in ocean circulation (e.g., the AMOC) might also play a
role. Ocean circulation did not directly influence the $DIN_{xs}$ in the water column, but it rather
redistributed the $DIN_{xs}$ over the basin in the NAtl. This mechanism requires confirmation based





on new data. Our study also shows that both human activities and climate variations together
exert a discernable impact on the decadal variations of $DIN_{xs}$ in the subsurface waters of the
NAtl.
Human activities may have begun to influence the concentrations and stoichiometry of
nutrients, at least in the western NAtl, and profound changes have been verified on the western
NPO (Kim et al., 2014a) and Mediterranean Sea (Moon et al., 2016). These findings indicate
global-scale changes in marine biogeochemistry, caused by human activities that are
simultaneously influencing carbon sequestration and greenhouse gas emission (e.g., $N_2O$) (Duce
et al., 2008). Continuing monitoring of changes in $DIN_{xs}$ in the NAtl are needed to determine
whether the levels have followed the recent decrease in AND, particularly in the USA. Such
external perturbations could also alter the close homeostasis of the marine N cycle and its
feedback to climate (Gruber and Deutsch, 2014).



**Supporting information**

 Detailed description of transects used in the analysis, and additional tables and figures.

**Author contribution**

 JYTY and KL designed the present work and drafted the manuscript. JYTY and JYM did the data analysis. JZZ, ISH, JSL, and EL contributed to discussion and interpretation of the data.

**Competing interests**

 We have no conflict of interest to declare.



## Acknowledgements

We wish to thank all of scientists who contributed to data used in this study. This work was supported by the National Institute of Fisheries Science (R2020044) and by "Management of Marine Organisms causing Ecological Disturbance and Harmful Effects" funded by the Ministry of Ocean and Fisheries (MOF). Additional support for JYTY was provided by "The Principal's Fund" of Xiamen University (ZK1114). JZZ was supported by NOAA Ocean and Atmospheric Research. The scientific results and conclusions, as well as any views or opinions expressed herein, are those of the authors and do not necessarily reflect the views of NOAA or the US Department of Commerce. This is the State Key Laboratory of Marine Environmental Science contribution NO. melpublication2020387.





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


**Figure captions**
**Figure 1.** (a) Nutrient sampling locations (black dots) in the North Atlantic Ocean (NAtl).
All datasets from the meridional (A22, A20 and A16N) and zonal (A05) cruises were
collected in the GLODAPv2 product and CLIVAR database (see text). The red dashed line
indicates the region of STMW formation, and the solid arrows indicates streamlines of high
transport in the western NAtl (i.e., the Gulf Stream, modified from Palter et al., 2005). The
color scale indicates the model-derived atmospheric $NO_x$ deposition into the NAtl for 2000
(Dentener et al., 2006).
**Figure 2.** The vertical distributions of $DIN_{xs}$ in the upper 1500 m for difference cruises along
three meridional transects are shown in, (a) A22, (b) A20, and (c) A16N, respectively. The
insets in (a−c) show the average rates (with 95% confidence limits) of $\Delta DIN_{xs}$ at 200−600 m
and 1200–1500 m averaged for each 3°−8° latitude interval between GO-SHIP (2010s) and
WOCE (late 1980s to 1990s) time periods along each transect (see Table S1).
**Figure 3.** Temporal trends of $DIN_{xs}$ anomalies (dots) for the corresponding latitude interval
for the subsurface potential density intervals $\sigma_\theta$ along the three meridional transects (a) A22,
(b) A20, and (c) A16N in the NAtl. Data from A05 obtained in 2010 at three crossover sites
are also shown (triangles). $mDIN_{xs}$ values in parentheses indicate the mean $DIN_{xs}$ in GO-
SHIP dataset. The selected $\sigma_\theta$ intervals are typically located at the depth intervals of 200−600
m with $DIN_{xs}$ maximum along each transect. Note that the selected $\sigma_\theta$ interval (= 27.2–27.6)
in the subpolar region along A16N (45°N−60°N) is different from that in the subtropical
region, as $\sigma_\theta$ for 200−600 m depth becomes larger in the high-latitude region. Besides repeat
cruises of these transects, the data sets from other cruises with similar tracks (Fig. 1) in the
sub-regions were included for comparison. The $DIN_{xs}$ values were corrected by the changes
in AOU to remove the contribution from remineralization of organic matter (see text). The
data points connected by the dashed lines indicate that the $\Delta DIN_{xs}$ were statistically



significant in these regions ($p < 0.05$, Student's t-test and ANOVA with Games-Howell test).
Otherwise, the data that were statistically unchanged are not connected by the dashed lines.
The smoothed trend using the 5-year rLowess filter for $DIN_{xs}$ anomalies of the STMW at the
same depth ranges of the BATS site (near A22) is also shown (the gray shading in a). The
gray dashed lines indicate $DIN_{xs}$ anomaly equals to zero.
**Figure 4.** As for Figure 3, except for DIP anomalies. mDIP values in parentheses indicate the
mean DIP in GO-SHIP dataset. The gray dashed lines indicate DIP anomaly equals to zero.
Date from the BATS site are also included in (a).
**Figure 5.** (a) The rates of $\Delta DIN_{xs}$ in the subsurface waters (200−600 m) along the four
transects between GO-SHIP and WOCE time periods (see Table S1). The study area is
divided into 10 subregions of 10° longitude by 5°−15° latitude along the transects A22
(boxes 1–2), A20 (boxes 3–5), part of A05 (box 6) and A16N (boxes 7–10). The statistically
significant changes (Student's t test and ANOVA with Games-Howell test, $p < 0.05$) are
marked with the superscript "t" for the box numbers. (b) Temporal variations of $DIN_{xs}$
anomalies (open dots and their fitting curve as black curve) in the western NAtl in which the
subsurface $DIN_{xs}$ increased significantly (boxes 1−3 in a). Trend in $DIN_{xs}$ anomalies in the
subsurface waters at the BATS site is shown in gray shading (the same as Fig. 3a). To ensure
consistent comparisons between atmospheric N deposition rates and seawater $DIN_{xs}$
anomalies, the seawater $DIN_{xs}$ anomaly values were shifted by approximately 15 years. The
15-year shift corresponded to the mean time period that had elapsed since a given subsurface
water mass had last been in contact with the atmosphere prior to subduction. The year that the
subsurface water mass in the NAtl last had contacted the atmosphere was calculated using the
CFCs contents in that subsurface water. The history of observed atmospheric wet deposition
(WD) of $NO_x$ from the US Atlantic coast. Orange curve and its shading show the 5-year
moving average values and the range of the 95% confidence intervals, respectively (the





monitoring sites are presented in Table S3). Blue curve indicates the $NO_x$ emission from the
USA. The $NO_x$ emission strongly correlates with the WD of $NO_x$ ($r = 0.93$, $p < 0.01$).



**Figure 1**

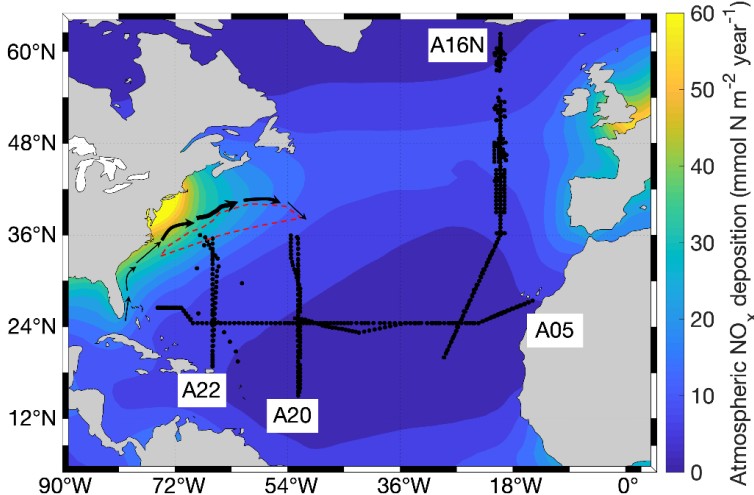






**Figure 2**

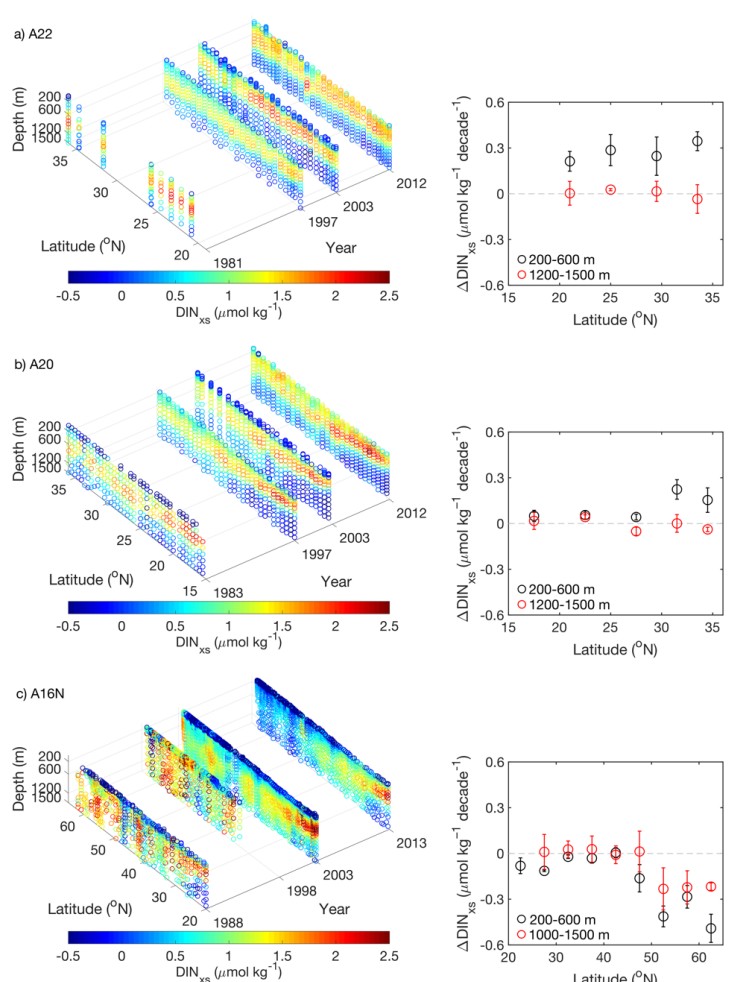






**Figure 3**

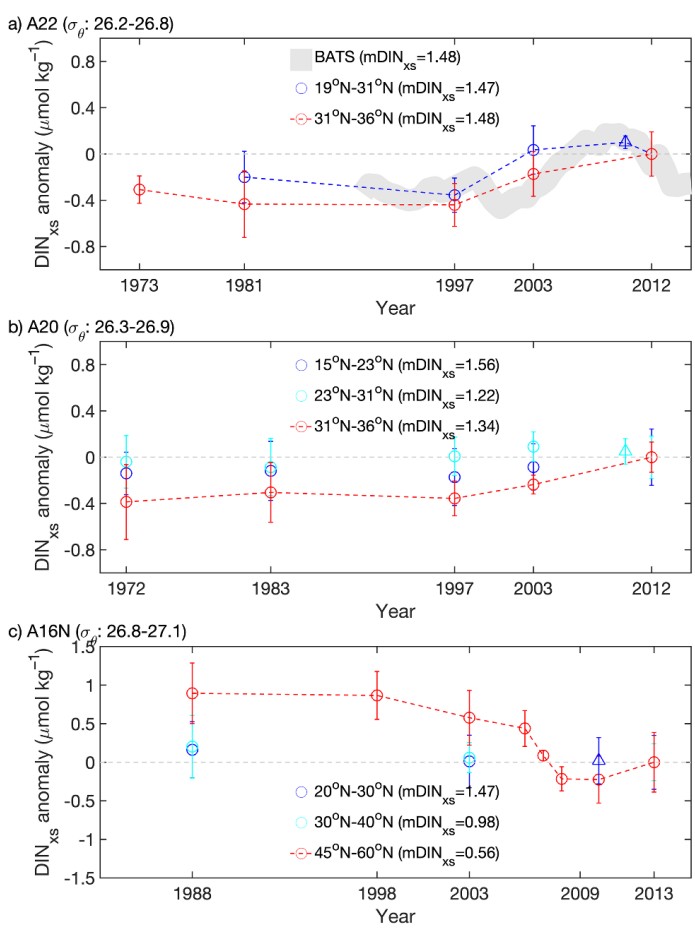






**Figure 4**

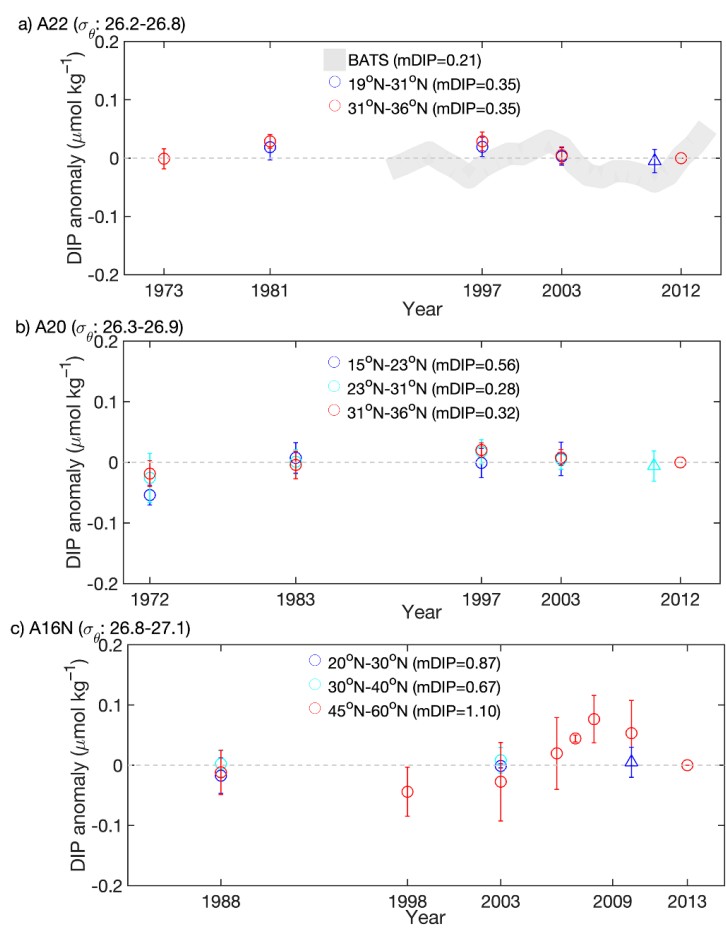




**Figure 5**

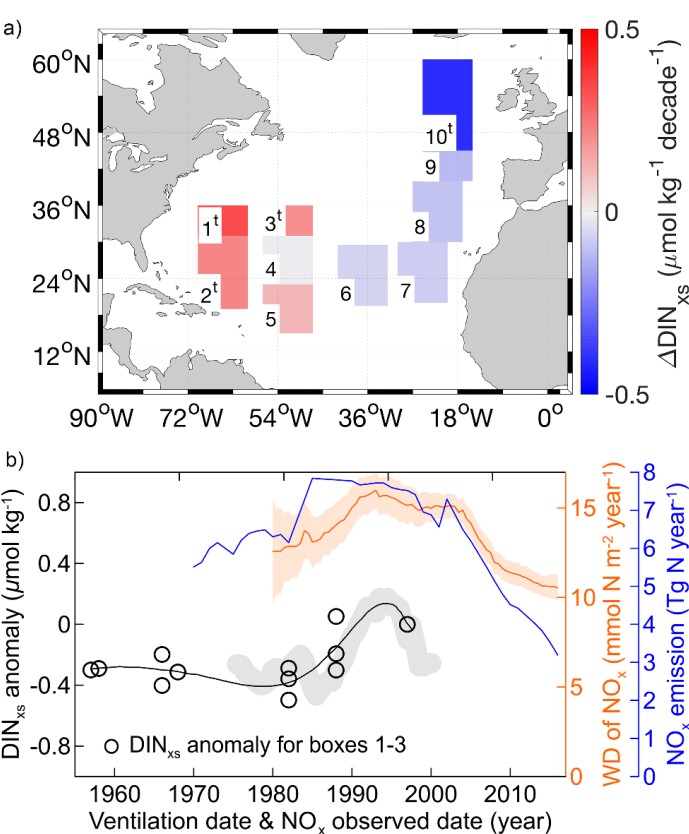

