# Peer review of "Contrasting decadal trends of subsurface excess nitrate in the western and eastern North Atlantic Ocean 4 5 Jin-Yu Terence Yang1,2, Kitack Lee1\*, Jia-Zhong Zhang3, Ji-Young Moon1, Joon-Soo Lee4, 6 In-Seong Han4, Eunil Lee"

_Biogeosciences, 2020_

## Referee Comment (RC1) · Anonymous Referee #1 · 19 Apr 2020

Review of 'Contrasting decadal trends of subsurface excess nitrate in the 2 western and eastern North Atlantic Ocean' by Yang et al.

This is a nice contribution that I recommend be published. I do have some comments that should be addressed and these are detailed below. Overall the manuscript is well written and the figures are clear and complete.

Line 45: 'an evidence' – change to just 'evidence'?

Please include a short discussion on the potential for any bias as a result of not having reliable concentration data <0.1 $\mu$mol kg–1 for DIN and 0.01 $\mu$mol kg–1 for DIP.

[Figure]

Lines 125ň–128: Please make it clearer whether this procedure was conducted by the authors of the current manuscript, or if this is a correction made prior to publication of the datasets the authors use. I also do not fully understand what this correction does? Please explain in clear terms why this correction need to be performed (i.e., why do the datasets need this correction to be made comparable in the first place?). Some details for this might be shifted from the supplement to the main text. An alternative option would be to state explicitly that this is discussed in more detail in the supporting information, but it would be useful if the key reason behind the corrections could be summarized succinctly in the main text.

Paragraph stating line 130: What is the cause of this inter-dataset difference? Analytical measurement errors?

Line 151: "In addition, the effect of seasonal variations on DINxs signals at this depth layer is generally insignificant," Please clarify, which layer are the authors referring too? Two different layers are discussed earlier in the paragraph.

Line 190: "Based on multiple cruises along each transect, changes in DINxs were discernable over the 191 decadal periods; these changes were most pronounced between 200 m and 600 m (Fig. 2)" How many data time points are these rate calculations based on? I understand this varies depending on the cruise line. I think it is important to include this information somehow on Figure 2.

Line 197: "Moreover, the $\Delta$DINxs values remained close to zero in the intermediate waters (1200–1500 m) in the western and eastern subtropical NAtl (Fig. 2). This observation confirms that the marked changes in DINxs largely occurred in the subsurface waters." This does not seem to be the case for the A16n line (i.e., deeper waters show the same trend as the surface waters here).

Line 203: "layer of the DINxs maximum decreased since 1997" Do the authors mean 'increased' instead of 'decreased'?
Paragraph starting line 220: Please attempt to describe N deposition rates quantitatively. i.e. to back up statements such as 'pronounced increase' and 'considerable AND input'

Paragraph starting line 234: Do the authors use the mean rate of deposition for the coastal AND sampling sites? Can an error bar therefore be added to the deposition trend in Fig. 5? This would help support the statement 'trend . . . commonly found at AND monitoring sites'

Line 264: ". . .although the mismatch between the observed time lag and the ventilation age of water masses may be due, at least in part, to the biological processes." For full clarity, please briefly specify the biological processing being referred to here.

Lines 274Âň–278: But here anthropogenic nutrient input is from a different continent? Please clarify.

Line 320–322: Would the detection limit of phosphate in surface waters be low enough to detect this change due to increased N2 fixation?

Line 429: "particularly in the USA" Rephrase to "particularly from the USA"?

Line 732: ". To ensure consistent comparisons between atmospheric N deposition rates and seawater DINxs anomalies, the seawater DINxs anomaly values were shifted by approximately 15 years." Please state exact time shift and if it was added or subtracted.

---

## Referee Comment (RC2) · Anonymous Referee #2 · 19 Apr 2020

The paper "Contrasting decadal trends of subsurface excess nitrate in the western and eastern North Atlantic Ocean" presents an impressive compilation of nutrient data from the North Atlantic (NA) Ocean. The main result is a DIN excess increase in subsurface waters of the western NA during the last 3 decades, which might be explained by anthropic atmospheric nitrogen deposition. Some other results, interesting interpretation and discussion follows, but according to my analysis the main result should be discussed before, relying on the accuracy of DIP measurements necessary for DIN excess calculation. I will begin with this major concern and follow with other comments.

Major concern:

Page 5 (106-109): the analysis of nutrient data was based only on concentrations greater than 0.1 $\mu$mol kg-1 for DIN and 0.01 $\mu$mol kg–1 for DIP. These concentration levels approximate the detection limits of DIN and DIP for the analytical methods used in the field observations (Zhang 109 et al., 2001; Hydes et al., 2010).

A similar colorimetric method is used to measure nitrate and phosphate in seawater using autoanalyzers, and there are no apparent reasons to consider a detection limit which is 10 times lower for phosphate than for nitrate. It would have been the case if the cell used to measure phosphate compare to nitrate had been 10 times longer, but it was not the case at least for the WOCE and previous cruises. Considering Redfield proportion (N:P=16:1) or what you have considered (N:P=15:1), it is clear that more than 10 times increased precision is useful to correctly interpret biogeochemical processes in the Sea. It is the reason why many efforts were done to lower the quantification limits for phosphate measurements, using nanomolar methods particularly in surface waters. A recent paper stipulates that "The underlying reason for a limited understanding in the distribution of surface DIP is that the standard methodology has high variance and low interlaboratory accuracy, below ~100 nM" (Aoyama et al., 2016; Martiny et al. 2019). Even if 100 nM may appear as a higher limit, it is the one you have considered for nitrate, and following my argument in the first sentence of this paragraph, it may be a plausible accuracy value. I ask you therefore to determine the uncertainties of DIN excess values, considering a 100 nM uncertainty in DIP measurements, in order to see if the increasing trend in DIN excess is still observable in this condition. This point is my major concern. It should at least be put forward first in your discussion.

Other comments:

Line 36: You use DIN for nitrogen; therefore, it would be preferable to use DIP for phosphate.

Lines 48-52 It would be preferable to reinforce the demonstration of the main result rather than propose new hypotheses far from the main result.

Line 55 Nr means nothing for me. You could use DIN, defined as the sum of nitrate $NO_3^-$, nitrite $NO_2^-$ and ammonium $NH_4^+$ where nitrite is usually negligible.

Lines 59-60 Anthropogenic nitrogen deposition (AND) to the contemporary ocean is comparable in magnitude to marine biological $N_2$ fixation: add reference(s).

Line 64 Delete pollutant nitrogen, replace with DIN.

Lines 93-94 The sentence at the end of this paragraph uses an older reference than the statement just before, making it a bit confusing.

Lines 102-109 This refers to my major concern explained in the upper part of this report.

Lines 114-115 Which is a major source region of anthropogenic nitrogen according to a model-derived atmospheric NOx deposition (Dentener et al., 2006; Fig. 1). It is not data but model-derived prediction. This information is important and needs to be added here and not only in the figure legend. If you want to look at a comparison between prediction and data in another context, I invite you to read this short interesting paper (Grüber, 2016).

Lines 138-140 The finding that the subsurface deltaDINxs signals were considerably greater than the detection limit of DIN is a strong indication that our data adjustments probably did not influence the temporal trend of DINxs. I agree with the adjustments, but deltaDINxs signals depend on DIN and DIP, and the accuracy will largely differ depending on what you choose as a quantification limit for DIP (see my first comment).

Line 145 Deficit? Don't you mean excess?

Line 166 Fig. S4. See Line 109 in the SM, after (b), A20 is missing.

Line 178 I am not sure that the introduction of anomalies here helps the readers. You will then compare anomalies and anomalies of excess, which have completely different uncertainties.

Lines 183-200 It is the main result which needs to be reinforced with a discussion on DINxs uncertainties. I wonder if a result/discussion part on DIN evolution rather than DINxs (depending on DIP concentration which may be harder to measure with enough accuracy) evolution will not be more straightforward and easier to publish.

Line 189 Fig. S6. See Line 131 in the SM, different and no difference.

Lines 201-202 Fig. 3 presents DINxs and not deltaDINxs.

Line 203 (measurement year)?

Lines 221-222 which is subject to considerable AND input from the North American continent. Add reference(s).

Line 222 Recent studies suggest that the reduced form of nitrogen... Do you mean $NH_4^+$?

Line 224 Thus? Marine nitrogen fixation is an autochthonous process which influences DINxs!

Line 225 Therefore?

Line 226 Emissions? Only deposition. If you add significant amount of $NH_4^+$ by the atmosphere, it will certainly influence DINxs values. I am not able to follow your reasoning there. NOx is introduced without being defined.

Lines 266-278 All this part will be clearer if DIN inventory is used instead of DINxs inventory.

Line 315 In this region (boxes 1–3). Please, add Fig. 5a.

Line 317" The atmospheric NOx deposition. Replace with "the modelled atmospheric NOx deposition".

Line 346 The formation of the STMW is generally enhanced when the NAO index becomes negative. Please, add reference(s).

Line 356 Fig. S7b. Please also represent the DIN anomaly, and for each graph use the proportion you have defined (N:P=15:1) in the axes to better represent N and P. Refer to my remark for Fig. S12.

Line 370 but little change in DIP was observed (Fig. S12). If you correctly represent the axes using N:P=15:1, as defined as a relevant ratio by yourself, you might not conclude that little change in DIP was observed. For DIN represented between 12.5 and 14.5 $\mu$mol kg-1, please represent DIP between 0.83 and 0.97 $\mu$mol kg-1. It is the only way to graphically compare N and P evolutions. Are the evolutions of DIN and DIP different? or are the uncertainties between DIN and DIP measurements, at the level needed, different?

Line 364-409 All this part is speculative. A paper focusing on proving the main result would be more interesting.

References cited:

Aoyama, M. Abad, C. Anstey, M. Ashraf, A. Bakir, S. Becker, S. Bell, M. Blum, R. Briggs, F. Caradec, F. Cariou, M. Church, L. Coppola, M. Crump, S. Curless, M. Dai, A. Daniel, E. de Santis Braga, M. E., 2016. Solis, J.-Z. Zhang, IOCCP-JAMSTEC 2015 Inter-laboratory calibration exercise of a certified reference material for nutrients in seawater.

Gruber, N. 2016. Elusive marine nitrogen fixation. Proc Natl Acad Sci USA 113(16):4246–4248.

Martiny, A.C., M. W. Lomas, W. Fu, P. W. Boyd, Y.-l. L. Chen, G. A. Cutter, M. J. Ellwood, K. Furuya, F. Hashihama, J. Kanda, D. M. Karl, T. Kodama, Q. P. Li, J. Ma, T. Moutin, E. M. S. Woodward, J. K. Moore, 2019. Biogeochemical controls of surface ocean phosphate. Sci. Adv. 5, eaax0341.

---

## Author Comment (AC1) · 22 May 2020

Review of 'Contrasting decadal trends of subsurface excess nitrate in the 2 western and eastern North Atlantic Ocean' by Yang et al.
This is a nice contribution that I recommend be published. I do have some comments that should be addressed and these are detailed below. Overall the manuscript is well written and the figures are clear and complete.
We thank Referee #1 for the positive evaluation and insightful comments. We have addressed the concerns raised by this referee in the revised manuscript, and thoroughly describe all changes made in our responses. Where no change to the text has been made we provide a full justification.

Line 45: 'an evidence' – change to just 'evidence'?
Please include a short discussion on the potential for any bias as a result of not having reliable concentration data <0.1 μmol kg$^{-1}$ for DIN and 0.01 μmol kg$^{-1}$ for DIP.
(Change made): A short discussion about possible biases in results has been added to the revised manuscript (lines 116–119). Our analysis for estimation of excess DIN focused exclusively on data collected from 200–600 m depth, where nutrient concentrations were greater than 1.4 μmol kg$^{-1}$ for DIN and 0.08 μmol kg$^{-1}$ for DIP. More explicitly, the lower ends of the DIN and DIP concentrations in these targeted waters are several-fold higher than the detection limits of DIN and DIP. As low DIP concentration (< 0.1 μmol kg$^{-1}$) may result in uncertainties (Martiny et al. 2019), to eliminate any potential bias in the DIN$_{xs}$ estimates we did not use those DIP and accompanying DIN data (accounting for 1.4% of the total 1955). Removal of the low DIP concentration data did not alter our finding (e.g., the trend of increasing excess nitrate in the western subtropical NAtl).

Lines 125–128: Please make it clearer whether this procedure was conducted by the authors of the current manuscript, or if this is a correction made prior to publication of the datasets the authors use. I also do not fully understand what this correction does? Please explain in clear terms why this correction need to be performed (i.e., why do the datasets need this correction to be made comparable in the first place?). Some details for this might be shifted from the supplement to the main text. An alternative option would be to state explicitly that this is discussed in more detail in the supporting information, but it would be useful if the key reason behind the corrections could be summarized succinctly in the main text.
(Explanation provided and changes made): We agree that we did not clearly describe data calibration issue in the original manuscript. In the revised manuscript (lines 136–143) we have included a section describing the data calibration and methods, which reads:
*The GLODAPv2 product includes data obtained from > 700 cruises during the period 1972–2013. These large datasets collected in different years and by different investigators may contain some systematic and analytical errors. To remove these systematic errors, quality control of the data was performed by Key et al. (2015) and by Olsen et al. (2016), largely based on comparison of repeated measurements made for waters deeper than 2000 m at the same locations. Any biases found were corrected by applying adjustment factors to the raw datasets, and the adjusted datasets were reported in the GLODAPv2 product (Key et al., 2015; Olsen et al., 2016).*

The corrections noted in italics above were performed by two groups (Key et al., 2015; Olsen et al., 2016). More recent data, collected during the 2010s, were not thoroughly compared with the data obtained in the 2000s or earlier. In addition, we removed the influence of remineralization of organic matter by considering the changes in AOU among different cruises. Therefore, we applied additional minor adjustments to all of the datasets obtained over three decades. In the absence of these small corrections (summarized in Table S2) we still found consistent increasing rates of excess DIN using the original uncorrected datasets (solid symbols); for example, since 1997 along the transect A22.

[Figure]

Paragraph stating line 130: What is the cause of this inter-dataset difference? Analytical measurement errors?
(Change made): In the revised manuscript (lines 136–138) we have stated that the difference in the N and P concentrations was largely caused by use of different analytical instruments and analysts.

Line 151: "In addition, the effect of seasonal variations on DINxs signals at this depth layer is generally insignificant," Please clarify, which layer are the authors referring to? Two different layers are discussed earlier in the paragraph.
(Change made): In the revised manuscript (lines 169) we have indicated that the effect of seasonal variations in DINxs signals found at 200–600 m depth was insignificant, because this depth range is typically deeper than the winter mixed layer in the study area.

Line 190: "Based on multiple cruises along each transect, changes in DINxs were discernable over the decadal periods; these changes were most pronounced between 200 m and 600 m (Fig. 2)" How many data time points are these rate calculations based on? I understand this varies depending on the cruise line. I think it is important to include this information somehow on Figure 2.
(Change made): Data from three cruises that occurred between the GO-SHIP and WOCE time periods were used to calculate the DINxs change in Figure 2. For the A22 transect we used 418 and 187 data points for the depth intervals 200–600 m and 1200–1500 m, respectively. For those depth intervals we used (respectively) 371 and 208 data points for the A20 transect, and 1168 and 613 data points for the A16N transect. We have added the number of data points used to the caption of Figure 2.

Line 197: "Moreover, the ΔDINxs values remained close to zero in the intermediate waters (1200–1500 m) in the western and eastern subtropical NAtl (Fig. 2). This observation

confirms that the marked changes in DINxs largely occurred in the subsurface waters." This does not seem to be the case for the A16n line (i.e., deeper waters show the same trend as the surface waters here).

(Change made): We agree that the DINxs changes in the subpolar region along the A16N transect occurred in both the subsurface and intermediate waters, whereas changes in the subtropical regions were only found in the subsurface waters. In the revised manuscript (lines 215–219) we have clarified that the smaller decrease in the $DIN_{xs}$ values in the intermediated waters north of 45°N along the A16N transect relative to those in the subsurface waters was probably associated with the deep winter convection and formation of the North Atlantic Deep Water in the subpolar NAtl. In this subpolar region there is a close connection between the subsurface and intermediate waters, which probably led to the DINxs decrease in the subsurface and intermediate waters. In contrast, changes in DINxs in the subtropical region were only found in the subsurface waters.

Line 203: "layer of the DINxs maximum decreased since 1997" Do the authors mean 'increased' instead of 'decreased'?

(Change made): We have changed "decreased" to "*increased*".

Paragraph starting line 220: Please attempt to describe N deposition rates quantitatively. i.e. to back up statements such as 'pronounced increase' and 'considerable AND input'

(Change made): Quantitative information on AND has been added to the main text (lines 241–243), which now reads: "*Model results have showed that the total AND over the NAtl basin in 2000 varied between 35–70 mmol N m$^{-2}$ year$^{-1}$, reaching higher values along the US coastal areas (Duce et al., 2008).*"

Paragraph starting line 234: Do the authors use the mean rate of deposition for the coastal AND sampling sites? Can an error bar therefore be added to the deposition trend in Fig. 5? This would help support the statement 'trend . . . commonly found at AND monitoring sites'

(Change made): The revised Figure 5 shows the 5-year moving means for wet $NO_x$ deposition (orange solid curve) along the coast of the USA. Note that the range of the 95% confidence intervals (indicated by the orange shading) indicates the error in those mean deposition values.

Line 264: ". . .although the mismatch between the observed time lag and the ventilation age of water masses may be due, at least in part, to the biological processes." For full clarity, please briefly specify the biological processing being referred to here.

(Change made): The biological processes involve planktonic assimilation of anthropogenic N during photosynthesis, and subsequent gravitational sinking and bacterial oxidation of organic matter at depth (lines 264–270). The oxidation of organic matter containing anthropogenic N may contribute to the elevation of the N:P signals at depth. In the revised manuscript (lines 285–289) we have briefly described these biological processes.

Lines 274–278: But here anthropogenic nutrient input is from a different continent? Please clarify.

(Change made): In the revised manuscript (lines 300–302) we have stated that the source of nutrient to the Mediterranean Sea is the "European continent". This is intended to indicate a similar phenomenon found in the other oceans, and highlights anthropogenic factors having a lag period of 20 years in their effect on subsurface nutrient dynamics.

Line 320–322: Would the detection limit of phosphate in surface waters be low enough to detect this change due to increased $N_2$ fixation?

(Change made): We agree that the change in surface DIP may be biased because of its very low concentration. We have rewritten this sentence (lines 347–350), which now reads: "*…if $N_2$ fixation activity had increased during the study period we would have expected more DIP in the surface ocean to be remineralized in the thermocline, leading to an increase in the subsurface concentration of DIP (Kim et al., 2014a), but no subsurface increase was observed (Fig. 4)*".

Line 429: "particularly in the USA" Rephrase to "particularly from the USA"?

(Change made): We have changed "particularly in the USA to "*particularly from the USA*".

Line 732: "To ensure consistent comparisons between atmospheric N deposition rates and seawater DINxs anomalies, the seawater DINxs anomaly values were shifted by approximately 15 years." Please state exact time shift and if it was added or subtracted.

(Change made): In the revised manuscript (line 738) we have stated that "*the seawater DIN$_{xs}$ anomaly values were shifted backward by 15 years.*"

---

## Author Comment (AC2) · 22 May 2020

The paper "Contrasting decadal trends of subsurface excess nitrate in the western and eastern North Atlantic Ocean" presents an impressive compilation of nutrient data from the North Atlantic (NA) Ocean. The main result is a DIN excess increase in subsurface waters of the western NA during the last 3 decades, which might be explained by anthropic atmospheric nitrogen deposition. Some other results, interesting interpretation and discussion follows, but according to my analysis the main result should be discussed before, relying on the accuracy of DIP measurements necessary for DIN excess calculation. I will begin with this major concern and follow with other comments.

[Response] We thank this referee for the constructive comments, and believe that the revised manuscript has been considerably improved as a result of the referee's suggestions. We have addressed all issues raised and provide our point-by-point replies below.

Major concern:
Page 5 (106-109): the analysis of nutrient data was based only on concentrations greater than 0.1 μmol kg-1 for DIN and 0.01 μmol kg–1 for DIP. These concentration levels approximate the detection limits of DIN and DIP for the analytical methods used in the field observations (Zhang 109 et al., 2001; Hydes et al., 2010).
A similar colorimetric method is used to measure nitrate and phosphate in seawater using autoanalyzers, and there are no apparent reasons to consider a detection limit which is 10 times lower for phosphate than for nitrate. It would have been the case if the cell used to measure phosphate compare to nitrate had been 10 times longer, but it was not the case at least for the WOCE and previous cruises. Considering Redfield proportion (N:P=16:1) or what you have considered (N:P=15:1), it is clear that more than 10 times increased precision is useful to correctly interpret biogeochemical processes in the Sea. It is the reason why many efforts were done to lower the quantification limits for phosphate measurements, using nanomolar methods particularly in surface waters. A recent paper stipulates that "The underlying reason for a limited understanding in the distribution of surface DIP is that the standard methodology has high variance and low interlaboratory accuracy, below ~100 nM" (Aoyama et al., 2016; Martiny et al. 2019). Even if 100 nM may appear as a higher limit, it is the one you have considered for nitrate, and following my argument in the first sentence of this paragraph, it may be a plausible accuracy value. I ask you therefore to determine the uncertainties of DIN excess values, considering a 100 nM uncertainty in DIP measurements, in order to see if the increasing trend in DIN excess is still observable in this condition. This point is my major concern. It should at least put forward first in your discussion.

(Explanation provided): Thank you for the insightful comments. First, we would like to clarify why the detection limits are different between DIN and DIP. Oceanographers have optimized their nutrient measurement methods for the range of nutrient concentrations in seawater. The WOCE and GO-SHIP repeat hydrography programs require full water column measurement of macronutrient concentrations in single analytical run using an autoanalyzer. The analytical methods are optimized to match the full range of DIN, DIP, and Si concentrations within their linear calibration range. As the maximum DIP concentrations in the water column were only 2–3 μM (approximately 1/15 of the nitrate concentration), the phosphate measurement method was optimized to the lowest calibration range to maximize

its sensitivity, resulting in the detection limit for DIP being approximately 10-fold lower than that for nitrate. Similar to the detection limits, the precision by the shipboard autoanalyzer was 0.1 µM for DIN and 0.01 µM for DIP. The overall uncertainty was 0.4% for both DIN and DIP in deep waters (GO-SHIP Cruise report: A16N, 2013).

(Change made): Our estimation of the excess N (DINxs) focused exclusively on data collected from 200–600 m depth, where the nutrient concentrations were greater than 1.4 µmol kg$^{-1}$ for DIN and 0.08 µmol kg$^{-1}$ for DIP. The lower ends of the DIN and DIP concentrations in these waters were several-fold greater than the detection limits for DIN and DIP. We understand the concern of this referee about the overall uncertainty of low DIP measurements, which include errors in both precision and accuracy. To be conservative and to eliminate any potential bias in the DIN$_{xs}$ estimates, in revised manuscript (lines 116–119) we have removed data (1.4% in a total of 1955) involving DIP concentration < 100 nM to. Removing these low DIP concentration data did not alter our finding of a trend of increase in excess nitrate in the western subtropical NAtl. We also accounted for errors in our DINxs estimates (the symbol size shown in Figures 3-5 covers the errors) by using an overall uncertainty of 0.4% for both DIN and DIP for the targeted water depth ranges. We have clarified this issue in the revised text (lines 165–168) and figure captions.

[Figure]

a) A22 ($\sigma_\theta$: 26.2-26.8)

Other comments:
Line 36: You use DIN for nitrogen; therefore, it would be preferable to use DIP for phosphate.
(Change made): We have use "*DIP*" for phosphate.

Lines 48-52 It would be preferable to reinforce the demonstration of the main result rather than propose new hypotheses far from the main result.
(Change made): To accommodate this suggestion, we have changed the last part of the Abstract and Section 3.5. The revised Abstract now reads: "*In contrast, a decreasing trend in subsurface DIN$_{xs}$ was observed after the 2000s in the eastern NAtl, particularly in the high latitudes. This finding was not associated with the comparable decrease in AND from Europe. Other natural processes (a possible decline in tropical N$_2$ fixation and weakening of the Atlantic meridional overturning circulation) may be responsible, but lack of more time-resolved data on N$_2$ fixation and meridional circulation is an impediment to assessment of these processes.*"

Line 55 Nr means nothing for me. You could use DIN, defined as the sum of nitrate NO3-, nitrite NO2- and ammonium NH4+ where nitrite is usually negligible.

(Change made): In the revised manuscript we have used bioavailable nitrogen (the sum of nitrate, nitrite and ammonium) throughout.

Lines 59-60 Anthropogenic nitrogen deposition (AND) to the contemporary ocean is comparable in magnitude to marine biological $N_2$ fixation: add reference(s).
(Change made): We have cited the publication of Duce et al. (2008) in the main text and included it in the reference list. This paper supports our statement above.

Duce, R. A., LaRoche, J., Altieri, K., et al.: Impacts of atmospheric anthropogenic nitrogen on the open ocean, Science, 320, 893-897, http://doi.org/10.1126/science.1150369, 2008

Line 64 Delete pollutant nitrogen, replace with DIN.
(Change made): We have replaced "pollutant nitrogen" with "*bioavailable nitrogen*".

Lines 93-94 The sentence at the end of this paragraph uses an older reference than the statement just before, making it a bit confusing.
(Change made): A more recent and relevant publication (Gruber et al., 2014) has been cited in the revised text and included it in the reference list.

Gruber, N., and Deutsch, C. A.: Redfield's evolving legacy, Nat. Geosci., 7, 853-855, http://doi.org/10.1038/ngeo2308, 2014.

Lines 102-109 This refers to my major concern explained in the upper part of this report.
(Change made): We have thoroughly addressed this concern in the revised manuscript. See our response to comment 1.

Lines 114-115 Which is a major source region of anthropogenic nitrogen according to a model-derived atmospheric NOx deposition (Dentener et al., 2006; Fig. 1). It is not data but model-derived prediction. This information is important and needs to be added here and not only in the figure legend. If you want to look at a comparison between prediction and data in another context, I invite you to read this short interesting paper (Grüber, 2016).
(Change made): In the revised manuscript (lines 123–125) we have explicitly indicated that the North America continent was identified as a major source of anthropogenic nitrogen, based on model predictions.

Lines 138-140 The finding that the subsurface deltaDINxs signals were considerably greater than the detection limit of DIN is a strong indication that our data adjustments probably did not influence the temporal trend of DINxs. I agree with the adjustments, but deltaDINxs signals depend on DIN and DIP, and the accuracy will largely differ depending on what you choose as a quantification limit for DIP (see my first comment).
(Change made): See our response to the major concern raised by this referee.

Line 145 Deficit? Don't you mean excess?
(Change made): To avoid confusion, we have deleted "deficit". The revised sentence now reads "*We calculated the DIN surplus relative to DIP...*".

Line 166 Fig. S4. See Line 109 in the SM, after (b), A20 is missing.
(Change made): We have added "*A20*" after (b).

Line 178 I am not sure that the introduction of anomalies here helps the readers. You will then compare anomalies and anomalies of excess, which have completely different uncertainties.

(Change made): To explain the anomalies in Figures 3 and 4 we have moved the sentence (*For each subregion, $DIN_{xs}$ (or DIP) anomalies indicates individual $DIN_{xs}$ (or DIP) values minus……*) to the captions to these figures.

Lines 183-200 It is the main result which needs to be reinforced with a discussion on DINxs uncertainties. I wonder if a result/discussion part on DIN evolution rather than DINxs (depending on DIP concentration which may be harder to measure with enough accuracy) evolution will not be more straightforward and easier to publish.

(Explanation only): For the evolution of DIN concentrations over time, changes in the DIN concentration can be associated with change in the rate of organic matter oxidation and the input of anthropogenic N at the target density range. The use of DINxs (excess DIN relative to DIP) was to remove the change in N associated with organic matter oxidation. By doing so, the evolution of DINxs can primarily be attributed to the evolution of input of anthropogenic N. We have also considered errors in the DINxs estimates in the revised manuscript. Please see our response to the major concern raised by this referee.

Line 189 Fig. S6. See Line 131 in the SM, different and no difference.

(Change made): As a west-east gradient in excess nitrate along the transect A05 at 24.5°N latitude in the central gyre was unclear, we have deleted this statement.

Lines 201-202 Fig. 3 presents DINxs and not deltaDINxs.

(Change made): We have changed "…$\Delta DIN_{xs}$ in the NAtl…." to "….*variations in $DIN_{xs}$ in the NAtl….*".

Line 203 (measurement year)?

(Change made): To avoid confusion we have replaced "measurement year" by "*the year of a cruise carried out*".

Lines 221-222 which is subject to considerable AND input from the North American continent. Add reference(s).

(Change made): A more relevant publication (Dentener et al., 2006) has been cited in the revised text and included in the reference list.

Dentener, F., et al.: Nitrogen and sulfur deposition on regional and global scales: A multimodel evaluation, Global Biogeochem. Cycles, 20, GB4003, http://doi.org/10.1029/2005gb002672, 2006.

Line 222 Recent studies suggest that the reduced form of nitrogen... Do you mean NH4+?

(Change made): In the revised manuscript we have amended the text to refer to two reduced forms of nitrogen (ammonium and dissolved organic nitrogen).

Line 224 Thus? Marine nitrogen fixation is an autochthonous process which influences DINxs!

(Change made): As the reduced form of nitrogen in atmospheric deposition largely originates from the surface ocean (Altieri et al., 2014; Altieri et al., 2016), we have replaced "this autochthonous reduced nitrogen…" by "*this marine-derived reduced nitrogen….*" By doing so, we have eliminated the possibility of $N_2$ fixation.

Line 225 Therefore?
(Change made): We have deleted "Therefore" in the revised text because the sentence reads well without it.

Line 226 Emissions? Only deposition. If you add significant amount of NH4+ by the atmosphere, it will certainly influence DINxs values. I am not able to follow your reasoning there. NOx is introduced without being defined.
(Change made): In the revised manuscript we have removed reference to NOx emissions from the sentence because these emissions did not influence seawater $DIN_{xs}$ values. In the revised text we have defined NOx as the oxidized form of nitrogen.

We agree that some atmospheric $NH_4^+$ is of anthropogenic origin, and to a minor extent could contribute to the $DIN_{xs}$ increase. Based on the observations of Altieri et al. (2014; 2016), only a small fraction of the reduced nitrogen (<30% of $NH_4^+$ and 17% of organic nitrogen) in atmospheric deposition is of anthropogenic origin.

Lines 266-278 All this part will be clearer if DIN inventory is used instead of DINxs inventory.
(Change made): As suggested by this referee, we have added the statement (lines 297–299): "*The atmospheric deposition possesses considerably high N:P ratio (up to >1000; Baker et al., 2010), which would mainly contribute to the DIN inventory in the western NAtl.*". This clarifies that the DIN inventory was used instead of the $DIN_{xs}$ inventory.

Line 315 In this region (boxes 1–3). Please, add Fig. 5a.
(Change made): We have added "Fig. 5a" next to "boxes 1-3".

Line 317" The atmospheric NOx deposition. Replace with "the modelled atmospheric NOx deposition".
(Change made): We have replaced "the atmospheric NOx deposition" with "*the modelled atmospheric NOx deposition*".

Line 346 The formation of the STMW is generally enhanced when the NAO index becomes negative. Please, add reference(s).
(Change made): The publication by Rodwell et al. (1999) has been newly cited to support the statement above.

Rodwell, M. J., Rowell, D. P., and Folland, C. K.: Oceanic forcing of the wintertime North Atlantic Oscillation and European climate, Nature, 398, 320-323, 1999.

Line 356 Fig. S7b. Please also represent the DIN anomaly, and for each graph use the proportion you have defined (N:P=15:1) in the axes to better represent N and P. Refer to my remark for Fig. S12.
(Change made): We have added the DIN anomaly and adjusted the axes accordingly.

Line 370 but little change in DIP was observed (Fig. S12). If you correctly represent the axes using N:P=15:1, as defined as a relevant ratio by yourself, you might not conclude that little change in DIP was observed. For DIN represented between 12.5 and 14.5 µmol kg-1, please represent DIP between 0.83 and 0.97 µmol kg-1. It is the only way to graphically compare N

and P evolutions. Are the evolutions of DIN and DIP different? or are the uncertainties between DIN and DIP measurements, at the level needed, different?

(Change made): As suggested by this referee, we have revised Fig. S12 to include comparable scales for DIN and DIP, using N:P=15:1. In this figure having comparable scales for N and P, the evolutions of DIN and DIP are different. Consequently, we rephrased the statement, which now reads: "*A significant decrease in the subsurface (300–500 m) DIN between 1998 and 2013 was also found at a site (68.0°N, 12.7°W) in the northern Iceland Sea, but concurrent decrease in DIP was not observed (Fig. S12). As a result, subsurface $DIN_{xs}$ therein declined remarkedly after 2005.*"

Line 364-409 All this part is speculative. A paper focusing on proving the main result would be more interesting.

(Change made): We have considerably shortened the speculative hypothesis and explanations in the revised Abstract and Section 3.5. In particular, we have deleted much of Section 3.5 (mostly speculative explanations for $N_2$ fixation and AMOC). This section retains a brief explanation for the decrease in anthropogenic N from Europe, as we evaluated this possibility based on data.